# Molecular Mechanisms of Ferroptosis and Its Role in Viral Pathogenesis

**DOI:** 10.3390/v15122373

**Published:** 2023-12-01

**Authors:** Riwei Huang, Jiang Wu, Yaodan Ma, Kai Kang

**Affiliations:** 1College of Coastal Agricultural Sciences, Guangdong Ocean University, Zhanjiang 524088, China; raywong0322@163.com (R.H.); wuj@gdou.edu.cn (J.W.); 2112104104@stu.gdou.edu.cn (Y.M.); 2College of Veterinary Medicine, South China Agricultural University, Guangzhou 510642, China

**Keywords:** ferroptosis, virus, mechanism

## Abstract

Ferroptosis is a novelty form of regulated cell death, and it is mainly characterized by iron accumulation and lipid peroxidation in the cells. Its underlying mechanism is related to the amino acid, iron, and lipid metabolisms. During viral infection, pathogenic microorganisms have evolved to interfere with ferroptosis, and ferroptosis is often manipulated by viruses to regulate host cell servicing for viral reproduction. Therefore, this review provides a comprehensive overview of the mechanisms underlying ferroptosis, elucidates the intricate signaling pathways involved, and explores the pivotal role of ferroptosis in the pathogenesis of viral infections. By enhancing our understanding of ferroptosis, novel therapeutic strategies can be devised to effectively prevent and treat diseases associated with this process. Furthermore, unraveling the developmental mechanisms through which viral infections exploit ferroptosis will facilitate development of innovative antiviral agents.

## 1. Introduction

Ferroptosis is a novel form of regulated cell death, relying primarily on intracellular iron to cause abnormal generation and degradation of intracellular reactive oxygen species (ROS), which deregulates the redox homeostasis mediating lipid-dependent oxidative damage [1]. The occurrence of ferroptosis is closely related to iron metabolism, amino acid metabolism and lipid metabolism [2,3,4]. Thus, its most prominent features are lipid peroxidation, ROS overproduction, and iron accumulation [5], but these can be inhibited by the antioxidant system such as glutathione peroxidase 4 (GPX4) and glutathione (GSH) [6]. Ferroptosis was first described as iron-dependent non-apoptotic cell death in 2012 [7], and since then, it has gradually become the focus of research. Recently, ferroptosis has attracted extensive attention in diseases, especially in tumors [8]. However, viral infection has been shown to be associated with cell death [9]; the specific mechanism is still unclear. In this review, we first briefly describe the major characteristic of regulated cell death. Next, we summarize the mechanism and regulatory network associated with ferroptosis, as well as its potential role in viral infection, while providing a perspective on unspecified problems related to ferroptosis and development of new antiviral drugs.

## 2. Inducers and Inhibitors of Ferroptosis

Ferroptosis is accompanied by increased intracellular iron, ROS, and lipid peroxidation levels. In addition, the ultrastructural morphological changes in cells undergoing ferroptosis, including the significant reduction in mitochondrial cristae, rupture of the outer mitochondrial membrane, and increased mitochondrial membrane density, distinguish ferroptosis from other regulated deaths [10]. At present, it has been found that a variety of compounds can induce or inhibit the occurrence of ferroptosis, called ferroptosis inducers or inhibitors (Table 1). The compounds can induce or inhibit the occurrence of ferroptosis through different mechanisms and targets, and now the inducers and inhibitors of ferroptosis are summarized, respectively.

## 3. The Mechanisms of Ferroptosis

### 3.1. Ferroptosis and Amino Acid Metabolism

Ferroptosis is closely related to amino acid metabolism [31] with glutamine and glutamate playing important roles during ferroptosis [37]. Under physiological conditions, the cystine/glutamate anti-transporter in cell membranes (System Xc^−^) ingests extracellular cystine, transporting it into the cell while releasing glutamate into the cytoplasm. At the same time, cystine is reduced to cysteine and used to synthesize GSH [38] (Figure 1). However, accumulating high glutamate concentrations in the nervous system produces toxic effects, inhibiting System Xc^−^ activity and inducing ferroptosis. Thus, glutamate-mediated toxic effects can be inhibited by ferroptosis inhibitors [7]. Glutamine is abundant in plasma and tissues in living organisms, where its degradation products are used as a raw material in numerous biochemical reactions. For example, glutamine is an intermediate of the tricarboxylic acid cycle, where its metabolism promotes ROS production by mitochondria. Thus, glutamine deficiency or degradation blocks inhibit ferroptosis. In addition, not all glutamine pathways induce ferroptosis, with GLS2 being the key pathway necessary for ferroptosis [39].

Glutathione is an essential factor in GPX4, containing an active hydrogen sulfur group (-SH) in its structure, which eliminates lipid peroxides and free radicals in the body through biochemical reactions [6]. Glutathione is synthesized by the combined action of glutamate–cysteine ligase (GCLM), glutathione synthase (GSS), and amino acids, including glutamate, cysteine, and glycine [40].

Glutathione peroxidases (GPXs) are selenium proteases abundant in the body. They are a central regulator of ferroptosis, mediating the reduction of lipid hydroperoxide [41]. For example, the knockout of the GPX4 gene in mice leads to cell death as ferroptosis [42]. GPX4 reduces ROS production using GSH to catalyze the reduction of hydrogen peroxide or hydroperoxides of phospholipids, which inhibits ferroptosis [43,44]. However, treatment with Erastin disturbs the biological synthesis of intracellular GSH by inhibiting cystine transport into the cell. This reduces GPX4 activity, leading to lipid ROS and hydroperoxide accumulation [45,46]. In addition RSL3, FIN56, and other ferroptosis inducers cause GPX4 degradation and lipid peroxidation, inducing ferroptosis [47,48,49] (Figure 1).

### 3.2. Ferroptosis and Iron Metabolism

Iron is an important microelement in living organisms stored in ferritin key to oxygen transport and many biochemical reactions and biosynthetic processes. It also regulates the accumulation of lipid peroxide and ferroptosis. Both iron deficiency and overload are harmful to the body; thus, iron ion balance is essential for homeostasis [50]. During the Fenton reaction, hydrogen peroxide (H_2_O_2_) is activated by ferrous ions (Fe^2+^), forming hydroxyl radicals (HO•).

Under normal physiological conditions, excess Fe^2+^ in the cell is transported by a chaperone, PCBP2 (poly (rC) binding protein 2). The Fe^2+^ is further excreted from the cell by the solute carrier family 11 A2/divalent metal transporter 1 (SLC11A2/DMT1) or the solute carrier family 40 members 1 (SLC40A1) and subsequently oxidized to ferric ions (Fe^3+^), a reaction catalyzed by ceruloplasmin (CP) or Hephaestin (HEPH), ferrous oxidases in the cell membrane [51] (Figure 2).

Ferritin (Ft) is a major intracellular iron storage protein complex containing a light chain (FTL) and heavy chain1 (FTH1) [52]. It binds and transports Fe^3+^ subsequently released out of the cell in the form of exosomes. However, few free intracellular Fe^2+^ form the cellular labile iron pool (LIP) during ferroptosis [53,54]. Ferrous ions react with phospholipid hydroperoxides (PLOOHs) during the Fenton chain reaction, forming HO•, Fe^3+^, and alkoxy radicals (PLO•). The three further react, producing phospholipid hydroperoxides (PLOO•) and ROS, leading to a destructive chain reaction and oxidative damage [55,56,57].

During ferroptosis, iron accumulation is the primary signal that initiates oxidative membrane damage. Iron accumulation leading to ferroptosis results from increased iron intake, inhibition of iron efflux, and iron storage [55,56,57]. To prevent iron accumulation, transferrin (Tf), a beta 1-globulin in serum, binds and transports the extracellular iron. Next, the transferrin receptor (TFR), a transmembrane glycoprotein in the cell membrane, interacts with transferrin, mediating the endocytosis of extracellular Fe^3+^. At lower pH conditions, Fe^3+^ ions are removed from transferrin and are rapidly reduced to Fe^2+^ by the six-transmembrane epithelial antigen of prostate 3 (STEAP3). The Fe^2+^ ions are released into the cytoplasm through SLC11A2/DMT1 and stored in LIP or PCBP2, causing an abnormal increase in intracellular iron, which mediates ferroptosis due to iron overload [58,59,60]. In addition, MYCN, a member of MYC (a proto-oncogene family), not only drives cell proliferation and apoptosis but also induces ferroptosis [61,62]. For example, MYCN upregulates TFR1 levels and induces TFR expression in neuroblastoma cells, promoting iron uptake, which causes iron overload. In addition, MYCN induces the inhibition of GPX4 in cells, causing cell sensitivity to ferroptosis [63]. At the same time, heat shock proteins beta-1 (HSPB1) influence the cellular sensitivity to ferroptosis by inducing phosphorylation mediated by protein kinase C (PKC), which downregulates Tf expression inhibiting TFR-mediated iron uptake, leading to ferroptosis [64]. In addition, heme oxygenase-1 (HO-1), a rate-limiting enzyme, is overactivated by a nuclear factor (erythroid-derived 2)-like 2 (Nrf2) during heme catabolism, mediating the degradation of heme to produce Fe^2+^, which induces ferroptosis [65,66] (Figure 2). On the contrary, HO-1 has also been reported to inhibit ferroptosis [67].

Dysregulated iron homeostasis through autophagy can positivity trigger ferroptosis [39]. Lysosomes release large amounts of iron in acidic and reductive environments as Fe^2+^ through the phagocytosis of ferritin and red blood cells, reducing the iron stores [68]. The nuclear receptor co-activator 4 (NCOA4) recruits ferritin to autophagosome for lysosomal degradation and iron release. Thus, NCOA4 mediates ferritin autophagy during ferroptosis, leading to cellular ferritin degradation and iron ion release, which causes an iron overload [69] (Figure 2).

### 3.3. Ferroptosis and Lipid Metabolism

Polyunsaturated fatty acids (PUFA) play an important role in ferroptosis. Phosphatidylethanolamine (PE) containing arachidonic acid (AA) or adrenic acid (AdA) is often the object of oxidation [70]. During ferroptosis, PUFAs containing arachidonic and adrenalin acids are activated. Subsequently, the activated PUFA binds to the coenzyme A (CoA), forming a long chain acyl-CoA under the action of ACSL4. Next, the long chain acyl-CoA is translocated into the phospholipids of the cell membrane, where the PUFA binds to the PE catalyzed by lysophosphatidylcholine acyltransferase 3 (LPCAT3) [71,72,73]. PE is oxidized when intracellular reactive oxygen species levels are elevated, and ferroptosis is induced when oxidation products accumulate to a certain level (Figure 3). And the protein kinase C beta II (PKC-beta II) activates ACSL4 and promotes ferroptosis by amplifying the lipid peroxidation levels [74]. In addition, the activity of PUFA is influenced by the competition of exogenous monounsaturated fatty acids (MUFA) during ferroptosis. It is more difficult to oxidize MUFA than PUFA due to the absence of diallyl. Therefore, cells use MUFA instead of PUFA, which prevents lipid peroxidation [75]. However, there are reports of MUFA inducing ferroptosis in mice with acute lung injury [76]. In addition, cells inhibit ferroptosis using phospholipase A2 (PLA2) to hydrolyze PUFA.

The acid sphingomyelinase (ASM)–ceramide pathway is a recently discovered ferroptosis signal pathway. Acid sphingomyelinase is the key enzyme in sphingolipid metabolism, catalyzing ceramide production in cells by hydrolyzing sphingolipids. It mediates the conversion of sphingomyelins to ceramides, resulting in the accumulation of ceramides in cells undergoing ferroptosis [77]. Ceramide is an intermediate signal molecule of sphingomyelin (SM) metabolism, regulating tissue apoptosis [78]. Under the action of Erastin, the activation of ASM inhibits System Xc^−^ and promotes glutathione depletion, Gpx4 autophagic degradation, and ROS production, leading to lipid peroxidation. In addition, ASM promotes the formation of autophagosomes in ferroptosis [79].

Lipoxygenases (LOXs) are non-heme-iron-containing fatty acid dioxygenase catalyzing the peroxidation of PUFA by phosphorylase kinase G2 (PKG2). Some LOXs also oxidize biofilms and lipoproteins [80,81]. LOXs oxidize PEs to their corresponding hydroperoxy derivative (LOOH), which is further converted into bioactive lipid mediators, generating lipid signaling molecules and the knockout of LOXs inhibits Erastin-induced ferroptosis [82,83,84]. At the same time, p53, a classic tumor suppressor gene that induces spermidine/spermine N1-acetyltransferase 1 (SAT1) expression [85], mediates the expression of arachidonate-15-lipoxygenase (ALOX15), triggering ferroptosis via PE oxidation [70]. p53 mediates arachidonate-12-lipoxygenase (ALOX12) expression through the p53-SLC7A11 axis. It also regulates SLC7A11 while releasing ALOX12, which oxidizes PE resulting in ferroptosis [86] (Figure 3).

NADPH-cytochrome P450 oxidoreductase (POR) and NADH-cytochrome b5 reductase (CYB5R1) are key enzymes present in the endoplasmic reticulum during ferroptosis [87,88]. The POR-deficient cells inhibit ferroptosis inducers such as Erastin, RSL3, and FIN56 [89]. In addition, POR and CYB5R1 transfer electrons from NADPH to oxygen, forming hydrogen peroxide, which undergoes a Fenton reaction in the presence of iron ions to form hydroxyl radicals. The hydroxyl radicals formed mediate the plasma membrane lipid peroxidation, leading to membrane damage during ferroptosis [90].

## 4. Regulatory Network of Ferroptosis

Iron accumulation and lipid peroxidation are the main signals that initiate oxidative damage on the plasma membrane during ferroptosis. Although numerous complexes that induce ferroptosis through different signaling pathways, it ultimately occurs due to the inhibition of System Xc^−^ or GPXs, in the transporter-dependent and enzyme-dependent pathways. Ferroptosis is initiated in the transport protein-dependent pathway by inhibiting cell membranes transport proteins such as cystine/glutamate anti-transporter and enzyme-dependent pathway by inhibiting intracellular antioxidant enzyme expression.

### 4.1. Inhibition of System Xc^−^ Induces Ferroptosis

System Xc^−^ consists of SLC7A11 and solute carrier family 3 member 2 (SLC3A2). It is a heterodimer crucial in intracellular glutathione synthesis by transporting cystine and glutamate, depending on the sodium ions concentration [91].

p53 is a tumor suppressor gene regulating ferroptosis by suppressing SLC7A11 transcription and upregulating Glutaminase 2 (GSL2) under high concentrations of ROS. *p53* targets SLC7A11 in System Xc^−^. Erastin upregulates p53 and downregulates SLC7A11, inhibiting the expression of System Xc^−^ and reducing GSH levels, leading to ferroptosis. This has been demonstrated by p53 mutants, such as p53^3KR^, which regulate the expression of SLC7A11 [92,93,94]. In addition, Rho-family GTPase 1 (RND1) binds and de-ubiquitinates p53, inhibiting p53 degradation, which regulates the p53-SLC7A11 pathway, promoting ferroptosis [95]. GSL2, a core member of the mitochondrial glutaminases, is also transcriptionally targeted by p53 [96]. p53 inhibits the antioxidant system and induces ferroptosis by promoting the expression of GSL2, which catalyzes the hydrolysis of glutamine to glutamate, decreasing glutathione (GSH) levels and increasing ROS levels [71,97]. 

Nuclear factor erythroid-2-related factor 2 (Nrf2), a transcription factor in the cap‘n’collar (CNC) family, is strictly regulated by the kelch-like ECH-associated protein 1 (Keap1) [98,99]. The Keap1/Nrf2 pathway is key in the induction of the antioxidant system response and regulation of ferroptosis in an organism. Nrf2 controls the body’s resistance to oxidative stress by regulating SLC7A11, GPX4, glutamate-cysteine ligases (GCL), subunit GCLC, and GCLM glutathione reductase (GR), and other downstream expressions regulating ferroptosis [67,100,101,102]. 

The tumor suppressor *BRCA1*-associated protein-1 (BAP1) is a deubiquitylating enzyme (DUB) that removes histone H_2_A mono-ubiquitination at position 119 in lysine (H2A-K119ub) [103,104]. Thus, BAP1 mediates ferroptosis by blocking cystine uptake, GSH synthesis and inhibiting the expression of H2A-K119ub on the SLC7A11 gene and transcription of its mRNA [105].

Activation transcription factor 3 (ATF3), a transcription factor in the ATF/CREB (cAMP response element-binging protein) family, also plays an important role in oxidative damage. For example, ATF3 independently bypasses the p53-binding SLC7A11 promoter, downregulating its transcriptional activity and expression, thereby inhibiting System Xc^−^ in Erastin-induced ferroptosis [106]. In addition, ATF3 regulates SLC7A11 transcription by interacting with p53, which accelerates ferroptosis [107,108].

Autophagy regulatory protein Beclin1 (BECN1) is a core component of the class III phosphatidylinositol 3-kinase (PI3K) complex [109]. BECN1 is phosphorylated by AMP-activated protein kinase (AMPK), then binding SLC7A11 directly blocking System Xc^−^, which leads to ferroptosis. Therefore, knockdown of BECN1 inhibits ferroptosis induced by Erastin, Sulfasalazine, and sorafenib [110].

### 4.2. Inhibition of GPX Activity Induces Ferroptosis

The glutathione peroxidase (GPXs) family consists of selenium-containing proteases digesting peroxides, including GPX1~GPX8 in organisms. GPX4 is a single-subunit protein in the mitochondria, cytoplasm, and nucleus [111], directly reducing phospholipid hydroperoxides and inhibiting ferroptosis. RSL3 covalently binds GPX4, rendering it inactive, which induces ferroptosis [45].

During GPX4 synthesis, selenium substitutes sulfur in cysteine forming selenocysteine (Sec) as the active part of GPX4. As a result, GPX4 expression requires the insertion of Sec into the UGA codon in the GPX4 mRNA sequence through a combined effect of the elongation factor, selenocysteine insertion sequence (secis), secis-binding protein 2 (SBP2), Sec synthase, and Sec-tRNA for recoding [41,112]. 

The Mevalonate pathway (MVA) is the key pathway for Isopentenyl pyrophosphate (IPP) synthesis through a series of biochemical reactions using acetyl coenzyme A (Acetyl CoA) [113]. However, the maturation of Sec-tRNA requires the transfer of isopentane groups from IPP to Sec-tRNA precursors [114]. Therefore, regulating the MVA pathway mediates the GPX synthesis while inhibiting the pathway, inducing ferroptosis.

Cysteine dioxygenase 1 (CDO1) is a non-heme metalloenzyme catalyzing the conversion of cysteine to taurine [115]. The *MYB* gene serves as the transcription factor regulating CDO1 expression. In addition, the CDO1 expression in gastric cancer cells restores cellular GSH levels while enhancing the mRNA and protein expression of GPX4, thereby inhibiting Erastin-induced ferroptosis [116]. Therefore, the *MYB*-CDO1-GPX4 axis is key in regulating ferroptosis.

### 4.3. Other Regulatory Networks Associated with Ferroptosis

#### 4.3.1. LKB1-AMPK-ACC-PUFA Pathway

AMPK regulates energy and glucose metabolism in eukaryotic organisms. It contains three subunits mainly activated by the upstream tumor suppressor gene *LKB1* [117]. In glucose deficiency, the body consumes ATP, increasing the AMP/ATP ratio while causing energy stress. At the same time, the stress signal stimulates LKB1 to mediate activate AMPK phosphorylation. Acetyl CoA carboxylase (ACC), an AMPK downstream protein, regulates PUFA synthesis. Precisely, ACC is phosphorylated by AMPK, inhibiting its activity, subsequently inhibiting PUFA synthesis and ultimately ferroptosis [118,119].

#### 4.3.2. NAD(P)H-FSP1-CoQ10 Pathway

Ubiquinone, known as coenzyme Q10 (CoQ10), usually exists in the mitochondria. Its reduced form is ubiquinol (CoQH2). NADPH catalyzes the reduction of CoQ10 to CoQH2 and captures lipid peroxides, preventing lipid peroxidation [120,121]. Thus, the ferroptosis suppressor protein (FSP1) present in the plasma membrane acts as an antioxidant regulator of ferroptosis, trapping lipid peroxides through the independent GPX4 using NADPH. In addition, the knockdown of the intracellular *FSP1* enhances RSL3-induced ferroptosis [122]. However, squalene synthase (SQS), an enzyme involved in cholesterol synthesis, inhibits CoQ10 and Sec-tRNA activity. Therefore, the activation of SQS mediated by FIN56 leads to GPX4 depletion and the inhibition of the NAD(P)H-FSP1-CoQ10 pathway, promoting ferroptosis [16].

## 5. The Role of Ferroptosis in the Pathogenesis of Viral Infections

Regulatory cell death is an important means for the body to maintain normal physiological activities and homeostasis. As a form of regulatory death, ferroptosis can regulate inflammatory cells and remove cancer cells in the development of the disease. For viruses to maintain their survival activities, they must be constantly multiplying. However, viruses are different from other microorganisms in that they must multiply in living cells and as far as possible to hijack the host metabolism and enzyme system for macromolecules and energy. Therefore, viruses have a complete set of replication cycles, that is, need to go through the process of adsorption, penetration, uncoating, biosynthesis, assembly and release to produce new generation of viruses. Therefore, ferroptosis is often manipulated by viruses to regulate host cell servicing for viral reproduction.

### 5.1. Transferrin Receptors Mediate Viral Entry Associated with Ferroptosis

Viruses infecting host cells first need to specifically adsorb to host cell surface receptors, and in this process viral attachment protein (VAP) interacts with the host cell surface receptors, mediating viral entry [123]. During virus penetration, some encapsulated viruses are internalized into cells by fusing with the host cell membrane via fusion proteins at neutral pH after adsorption. Most enveloped viruses and naked viruses form endosomes by endocytosis upon adsorption and subsequently traffic to low-pH compartments, thereby mediating viral entry into the cytoplasm in the action of H^+^.

Many membrane proteins participate in intracellular trafficking events between compartments and the cell surface. TfR1, a membrane protein and also known as cluster of differentiation 71 (CD71), is endocytosed into the cell via clathrin-mediated endocytosis (CME) after interacting with Tf [124]. Subsequently, under the action of endosomal recycling and multiple factors, part of TfR1 is degraded after endosomal sorting, and another part of TfR1 returns to the plasma membrane to complete the recycle of the Tf-TfR1 complex [125]. It has been documented that multiple viruses not only may utilize TfR1 as a receptor but may also hijack the TfR1 transport partway to endocytose into the host cells and upregulate its expression. These may indirectly cause an increase in iron uptake, and ultimately lead to ferroptosis due to iron deposition, thereby exacerbating the progression of the disease in the later stages of viral infection. Wang et al. identified that TfR1 interacts with the Rabies virus (RABV) glycoprotein (GP) through its ectodomains and mediates RABV entry. In this process, RABV may hijack the TfR1 transport pathway co-transporting with metabotropic glutamate receptor 2 (mGluR2), a viral receptor of RABV, to endocytose into the cell via CME [126]. New World Arenaviruses (NWAV) are also similar to RABV when entering host cells. Nicolás et al. clearly indicated that the TfR1 apical domain interacts with NWAV GP1 [127]. Subsequently NWAV may be endocytosed into cells via TfR1-dependent or TfR1-independent CME and then virus–cell membrane fusion occurs under the action of NWAV GP2 and SSP in acidified endosomes [128,129]. TfR1 serves as a gate for SARS-CoV-2 infection. First, the SARS-CoV-2 spike protein binds to the specific receptor and then interacts with mGluR2. It subsequently migrates to the TfR1-containing clathrin coated pit (CCP) and interacts with TfR1 though spike protein and mGluR2, which hijacks the TfR1 transport pathway to enter into host cells [126]. Sokolov et al. found that Ferristatin II, a transferrin receptor inhibitor, inhibits SARS-CoV-2 replication through blocking the TfR1-mediated SARS-CoV-2 entry in vitro [130]. In porcine epidemic diarrhea virus (PEDV), as a coronavirus like SARS-CoV-2, TfR1 plays an active role in its internalization of PEDV. PEDV spike protein 1 interacts with the extracellular domain of TfR1. Then, it activates TfR1 tyrosine phosphorylation mediated by Src kinase and enhances TfR1 internalization to promote PEDV entry [131]. Mazel Sanchez et al. proposed the “revolving door” model, revealing that Influenza A virus (IAV) utilizes the TfR1 cyclic trafficking mechanism to endocytose host cells. Before endocytosis, IAV may bind to host surface receptors via HA and subsequently rolls laterally through the cell surface in search of a suitable entry site [132]. Martin et al. identified that TfR1 is an entry factor for HCV after interacting with CD81 and it may be required for HCV endocytosis but not for cell-to-cell infection [133]. The above evidence suggests that TfR1 and its traffic mechanism are essential for the entry of some viruses, but TfR1 is not the only receptor for viral adsorption and internalization. Most viruses use multiple receptors to complete the first phase of viral replication, and then subsequently internalize into the cell through different uptake pathways. Therefore, the development of inhibitory drugs against viral receptors such as TfR1 and mGluR2 or the use of these receptors as antiviral targets is necessary.

### 5.2. Potential for Virus-Mediated Ferroptosis to Facilitate Viral Release

Viruses need to be assembled and released outside the cell after using the host cell to complete genome transcription and synthesis of nucleic acids and proteins, then infect the next host cell. Enveloped viruses are mostly released in the form of budding or cytotoxicity. In the case of non-enveloped viruses, assembly often requires host cell lysis before virions can be released, and cell death may also be involved in this process. It has been found that the norovirus, a non-encapsulated virus, can trigger programmed cell death via the N-terminal structural domain of the nonstructural protein NS3 by disrupting the mitochondrial membrane structure [134]. Although ferroptosis is not as controllable as programmed death, late in some viral infections, pathogens may induce ferroptosis through the preinfection-mediated accumulation of ROS, Fe^2+^, and lipid peroxides or through cell death signaling pathways that in turn promote progeny virion release. However, this remains to be proven experimentally and more research is still needed to explain the underlying mechanisms.

### 5.3. Viruses Trigger Ferroptosis to Escape the Immune Response and Promote Viral Proliferation

Gpx4 plays an important role in antiviral immunization; as a key molecule in the fight against ferroptosis, it may ensure the normal activation of immune signaling pathways and the normal function of immune cells by maintaining redox homeostasis. Nevertheless, viruses have evolved a variety of strategies to evade the immune response in order to facilitate viral replication in host cells after years of struggling with the host immune system. Certain viruses can promote its replication by inducing immune cell ferroptosis or altering the host cell redox environment to fight against the immune system such as inhibiting or utilizing Gpx4, and altering host metabolism.

#### 5.3.1. GPX4 Provides a Stable Oxidation–Reduction State for Immune System

As an important molecule against ferroptosis, GPX4 provides a stable redox environment for the immune cells. MATSUSHITA et al. have found that the absence of Gpx4 can lead to ferroptosis and a reduction in T cells induced by lipid peroxidation [135]. Jia et al. found that treatment of primary mouse peritoneal macrophages with the GPX4 inhibitor RSL3 may induce macrophage lipid peroxidation. And this can significantly inhibit IFN-β protein expression and signaling in the DNA-sensing pathway induced by herpes simplex virus-1 (HSV-1) [136]. The above evidence suggests that Gpx4 is required for normal immune cell activity and activation of antiviral immune signaling pathways during viral infestation (Figure 4).

#### 5.3.2. Virus Targeting System Xc^−^-GSH-GPX4 Axis Induces Ferroptosis in Immune Cells

In order to survive, viruses have to evolve a variety of immune escapes to counteract or interfere with the host immune system, including the ability of viruses in persistent infections by altering the homeostasis to induce immune cell death. Hu et al. found that the exosome miR-142-3p secreted by HBV-infected hepatocellular carcinoma (HCC) cells could regulate the mRNA transcription level of SLC3A2, a subunit of system Xc^−^. In this study, the content of GSH decreased significantly through the detection, and the decreased GSH will cause inhibition of Gpx4 activity thereby promoting ferroptosis in M1-type macrophages and HCC cell proliferation [137]. Wang et al. found that the mRNA concentration of Gpx4 was significantly reduced after infection of Vero cells with SARS-CoV-2 thus increasing the risk of ferroptosis [138], and the excessive accumulation of mitochondrial ROS caused by the inhibition of Gpx4 promotes the glycolysis, which inhibits the T cells by altering the metabolism and thus promotes the replication of SARS-CoV-2 [139]. Thus, viruses can directly target the System Xc^−^-GSH-GPX4 axis and downregulate the mRNA levels of GPX4 or SLC7A11 through transcriptional regulation, which in turn induces ferroptosis and suppresses the immune response. (Figure 4).

#### 5.3.3. Viruses Regulate System Xc^−^ and GPX4 to Alter Redox Environment in Infected Cells

The System Xc^−^-GSH-GPX4 axis is an important pathway for the body to regulate peroxide levels. However, viruses could inhibit the expression of GPX4 and the subunits of System Xc^−^ in infected cells, thereby causing elevated peroxide levels and promoting ferroptosis in the service of viral reproduction and further deepening of disease progression. Cheng et al. showed that the expression of SLC3A2, SLC7A11 and Gpx4 are downregulated during the infection of Swine influenza virus (SIV), and the study also found that SIV-induced ferroptosis promoted its replication in A549 cells [140]. Liu et al. found that hepatitis B virus X protein (HBx) could stabilize the enhancer of zeste homolog 2 (EZH2) and promotes trimethylation of Lys-27 in histone 3 (H3K27me3), thereby inhibiting the expression of SLC7A11 in primary hepatocytes. This could affect the redox homeostasis stabilizing effect of Gpx4 and promoting ferroptosis in hepatocytes [141]. NDV as a tumor-lysing virus can selectively replicate in tumor cells and kill tumor cells. A study found that NDV upregulates p53 expression and downregulates SLC7A11 and Gpx4 expression during infection, inducing ferroptosis in tumor cells by causing a significant increase in the levels of ROS and peroxides [142]. In conclusion, viruses downregulate System Xc^−^ and GPX4 during infection through epigenetic or anti-oncogene regulation, leading to ferroptosis in infected cells, which promotes viral release or againsts tumor cells (Figure 4).

#### 5.3.4. Virus Targeting Keap1-Nrf2 Axis Affects Downstream GPX4 Activity

The Keap1-Nrf2 axis is an important signaling pathway in the body’s resistance to oxidative stress. Keap1 binds to and promotes the degradation of Nrf2 in the resting state. When oxidative stress occurs, keap1 dissociates from Nrf2, entering the nucleus and binding to the Antioxidant Response Element (ARE) to activate downstream genes to regulate the expression of antioxidant enzymes, including Gpx4 [143]. Thus, Nrf2 acts as an upstream regulator of Gpx4; some viruses can inhibit it by enhancing cellular oxidative stress to promote ferroptosis. Xu et al. found that HSV-1 enhances the action of Keap1 on Nrf2 to promote Nrf2’s ubiquitination and degradation, leading to reduction in the gene expression of antioxidant stress, which disrupts cellular homeostasis and promotes ferroptosis [144]. Liu et al. found that Nrf2 expression was significantly reduced during H1N1 infection and induced ferroptosis in human nasal epithelial progenitor cells (hNECs) through elevated intracellular oxidative levels caused by the NRF2-KEAP1-GCLC axis, resulting in nasal mucosal epithelial inflammation [145].

However, not all viruses induce ferroptosis in infected cells. Instead, the Epstein–Barr virus (EBV) can reduce the expression of Keap1 through the p62-Keap1-Nrf2 pathway in nasopharyngeal carcinoma cells (NPC), resulting in an increase in Nrf2 in the nucleus and a further upregulation of GPX4 expression. And high levels of Gpx4 inhibits ferroptosis in NPC, thereby promoting tumor growth [146]. In addition, Activating Transcription Factor 4 (ATF4) acts as an important regulator of endoplasmic reticulum stress during the development of HCC induced by Hepatitis C virus (HCV) and HBV, and He et al. revealed that ATF4 synergizes with Nrf2 to induce the expression of SLC7A11, which in turn regulates the function of Gpx4 and the synthesis of GSH to inhibit ferroptosis [147]. Gao et al. found that YAP/TAZ, a transcriptional effector of the Hippo signaling pathway, maintained the stability of ATF4 and facilitated its entry into the nucleus, synergistically inducing the expression of SLC7A11 [148]. In general, viruses regulate the process of ferroptosis to serve its own reproduction not only act on the System Xc^−^-GSH-GPX4 axis, but also regulate GPX4 through upstream signaling (Figure 4).

Viruses are fragile and susceptible to attack by the host’s immune system during the pre-infection period, when the viral load has not yet reached the logarithmic phase. While GPX4 plays a crucial role for the normal activity of the host immune system, viruses usually downregulate GPX4 by interfering with or affecting the activity of System Xc^−^ and GPX4 as well as SLC7A11 transcriptional regulation targeting the SLC7A11-GSH-GPX4 axis, which leads to pathogen multiplication. For late infection and virus-induced diseases, especially virus-associated tumor diseases, GPX4 is often upregulated to resist the cytocidal effect of host or drugs on tumor cells.

### 5.4. Viral Infection Alters Host Iron Metabolism

Iron in the organism is widely involved in and regulates biological processes; thus, iron homeostasis is crucial for life activities. Current studies have shown that iron is required for the proliferation of immune cells such as macrophages, T-cells and B-cells [149]. Iron is also required for immune cells to exert their effects, and in the oxygen-dependent bactericidal process of phagocytosis, iron is involved in causing the production of peroxides to destroy pathogenic microorganisms [150]. However, the biosynthesis and replication of nucleic acids and proteins during viral infections such as cytomegalovirus, human immunodeficiency virus (HIV), and HSV-1 also require iron [151], which would lead to iron imbalance caused by a competitive relationship between the host and pathogen for iron uptake.

#### 5.4.1. Viral Upregulates Intracellular Iron Levels and Promotes Viral Reproduction

During viral infection, viruses cause iron overload by directly or indirectly regulating the biosynthesis of transferrin and its receptor, ferritin and hepcidin. Excessive iron free or deposited throughout the body can be induced through the Fenton reaction, which induces an increase in the level of intracellular free radicals and ROS and promotes the ferroptosis process, thereby promoting viral infection. Hepcidin is a peptide hormone regulator secreted by liver cells, which mainly binds to and promotes the internalization and degradation of iron transport protein 1 (FPN1) on the surfaces of the duodenum, hepatocytes, and the reticuloendothelial system, thereby decreasing the absorption of iron by the epithelial cells of the intestine as well as the release of iron from macrophages and hepatocytes, ultimately resulting in a decrease in plasma iron levels [152]. Generally, hepcidin responds to the immune response to inflammation or infection and regulates iron metabolism in order to prevent iron overload or iron deficiency [153]. Chang et al. found that the mRNA levels of TFR-1 increase during HIV infection, which leads to increased iron uptake and elevated intracellular iron levels, and that elevated iron levels facilitate HIV transcription, replication, and release [154]. It has also been shown that the HIV-1 envelope glycoprotein gp120 and HIV-1-encoded transcriptional activator of transcription (TAT) induce endolysosome deacidification. This may cause iron efflux, leading to increased mitochondrial iron and ROS levels and thus increased cellular sensitivity to ferroptosis [155,156]. Lu et al. have shown that CVB3 can cause iron overload in host cells and the nuclear transcription factor Sp1 can recruit TFR into the nucleus and upregulate its expression, resulting in the increased activity of TFR 1 [157].Then, iron inflow into cells leads to iron overload and promotes ferroptosis. A study showed that SARS-CoV-2 can induce the host immune response and causes IL-6 hyperactivation [158]. And IL-6 has been shown in previous studies to promote hepatic transferrin uptake as well as to induce the synthesis of hepcidin and ferritin [159,160]. During SARS-CoV-2 infection, it was found that elevated ferritin levels activate NCOA4-mediated ferritin autophagy [161], whereas elevated ferritin levels inhibit the regulation of iron homeostasis by transferrin and macrophages, resulting in reduced iron efflux. Thus, the overexpression of IL-6 will lead to intracellular iron overload and thus ferroptosis. In addition, NDV also reduces ferritin through NCOA4-mediated ferritin autophagy to cause iron overload and induces ferroptosis in glioma cells [142].

The liver is an important organ in the maintenance of iron homeostasis, not only as a site for the synthesis of transferrin and a reservoir of iron, but also participating in the iron metabolism by synthesizing and releasing hepcidin. Hepatic iron deposition is observed in most patients infected with HCV [162]: although there is still no evidence directly indicating that HCV infection causes cellular iron death, a study has found that HCV infection causes the downregulation of hepcidin and increased serum ferritin concentrations in patients with chronic liver disease [163]. In another study, serum ferritin levels were found to show a trend of increasing and then decreasing in correlation with viral load levels, and the high expression during acute infection may be due to the ability of HCV to regulate intracellular iron content through ferritin, thereby enhancing HCV proliferation [164]. It has also been found that the non-structural protein of HCV, NS5A, inhibits the transcription of hepcidin [165]. Therefore, chronic HCV infection can cause excessive iron deposition in the liver and lead to iron overload, thereby increasing the risk of hepatic ferroptosis exacerbating liver injury. WANG et al. found that iron overload-induced liver injury could be rescued by ferrostatin-1, an inhibitor of ferroptosis [166], confirming that iron overload caused by HCV infection is highly associated with ferroptosis. In conclusion, iron is essential for the replication of the virus itself during viral infection because the virus is able to alter intracellular iron metabolism in the host cell by several means, thereby serving viral reproduction, and virus-induced iron overload further induces ferroptosis in the host cell (Figure 4).

#### 5.4.2. Viral Causes Decreased Intracellular Iron Levels to Promote Disease Progression

Not all viral infections result in host iron overload and promote ferroptosis. Ferroptosis is important in the suppression of cancer, and thus, some viral-induced cancers can inhibit ferroptosis and promote cancer cell growth by downregulating host iron levels through viral action. HBV is a double-stranded DNA virus that is also responsible for liver cancer and cirrhosis. Liu et al. found that HBV inhibits ferritin production through serine/arginine-rich splicing factor 2 (SRSF2) -mediated aberrant splicing of proliferating cell nuclear antigen clamp-associated factor (PCLAF) [167]. Wang et al. found that HBx-induced heat shock protein family A member 8 (HSPA8) stimulated HBV replication and inhibited ferritin production, and promoted the activation of hepatic stellate cells (HSCs) to induce liver fibrosis [168]. Zhang et al. found that miR-222, an exosome secreted by HBV-infected hepatocytes inhibits TFR expression [169]. The above evidence suggests that HBV can inhibit ferroptosis and promote the development of HCC and liver fibrosis by lowering iron levels, and therefore the treatment of HCC and liver fibrosis should be complemented by the addition of ferroptosis activators as appropriate (Figure 4).

### 5.5. Viral Infection Alters Host Lipid Metabolism Levels

Cells produce byproducts like ROS during metabolic processes, but the organism maintains redox homeostasis with GPX4 and some free radical trapping antioxidants such as CoQ10H2, thus avoiding the formation of phospholipid peroxides and thus preventing the occurrence of ferroptosis. However, some viruses are able to alter lipid metabolism to promote reproduction or to cause lipid accumulation, thereby increasing the risk of ferroptosis.

#### 5.5.1. Viruses Mediate Ferroptosis through ACSL Family Upregulation

ACSL4 as a key component of ferroptosis, catalyzes the activation of fatty acids by adding CoA into PUFAs to promote further lipid peroxidation production. Kung et al. found that ACSL4 is essential for the replication of enteroviruses such as coxsackie viruses, corona viruses, and influenza viruses, and the viruses promote the formation of viral replicative organelles and viral assembly through the recruitment of ACSL4, as well as the promotion of lipid peroxidation that induces ferroptosis in infected cells in order to support viral release [170]. Kannan Muthukumar et al. utilized in vitro experiments in mouse primary microglia (mPM) after exposure to HIV-1-encoded TAT, and found that TAT upregulates ACSL4 expression, mediating increased lipid peroxidation via the miR-204-ACSL4 axis [171] (Figure 4). In addition, ACSL1 as an ACSL family member like ACSL4 is involved in lipid metabolism and promotes the linolenic acid-induced ferroptosis. Xia et al. found that mouse hepatitis virus strain A59 (MHV-A59) induces ferroptosis in primary macrophages and promotes the formation of viral syncytia through the upregulation of ACSL1 expression, thereby facilitating virus reproduction [172].

#### 5.5.2. Virus Causes Persistent Infection through Ferroptosis

Ferroptosis can limit HCV replication, but this limitation can evade the host immune response and cause persistent infection. Yamane et al. found that the fatty acid desaturase FADS2 promotes iron-mediated lipid peroxidation through HUFA synthesis, causing conformational changes in the protein structural domains of the HCV replicase complex, thereby restricting HCV replication and causing persistent viral infections [173] (Figure 4).

In order to obtain benefits from infected cells, viruses promote viral replication and release and serve their own reproduction by regulating GPX4 activity, targeting the System Xc^−^-GSH-GPX4 axis to disrupt redox homeostasis, altering host iron metabolism and lipid metabolism. When the accumulation of intracellular lipid peroxides in the host cell reaches a certain level that exceeds the organism’s ability to remove superoxide or when divalent iron is in excess, ferroptosis occurs and further promotes the development of the disease. In addition, viruses can induce ferroptosis in immune cells, thereby disrupting the host immune response.

## 6. Conclusions

The host immune system’s fight against pathogenic microorganisms is a complex process that involves ferroptosis. Ferroptosis plays different roles in different pathogenic situations. In the interaction between pathogenic microorganisms and the host, ferroptosis is often manipulated by viruses to promote viral proliferation. Certain viruses such as RBV, NWAVs, and SARS-CoV-2 need to enter the cell through the transport pathway of TfR1, during which the viruses inhibit the degradation of TfR1 and upregulate the expression of TfR1 to promote the internalization of more viruses in the pre-replicative stage of viruses. The TfR1 upregulation results in the entry of more iron into the cell. Iron is not only beneficial to host cells, but also to viral replication, as certain viral replicative enzymes require iron to promote viral proliferation. With the accumulation of iron, some viruses further induce ferroptosis by downregulating host GPX4 and promoting the accumulation of lipid peroxides, which may be a means for viruses to inhibit immune cells from exercising immune escape or to release viral progeny, thereby promoting viral replication. For the later stages of viral infection, certain viruses promote their pathogenesis by downregulating ferroptosis, especially for oncogenic viruses, as ferroptosis is often detrimental to tumors. Therefore, for different periods of viral infection, ferroptosis inhibitors or inducers can be applied clinically to treatment.

Since the word ferroptosis was introduced in 2012, many scholars and researchers have conducted numerous studies on ferroptosis and its related fields. Although ferroptosis is mainly characterized by iron accumulation and lipid peroxidation, the mechanism and signaling pathways involved in ferroptosis in pathogenic microbe-host struggles are still unclear compared to other RCDs, and in depth studies are needed to provide more evidence and details to pave the way for the antiviral drugs or vaccines development. Thus, in the near future, unspecified issues related to ferroptosis will be resolved and new antiviral drugs and therapeutic approaches will be developed.

## Figures and Tables

**Figure 1 viruses-15-02373-f001:**
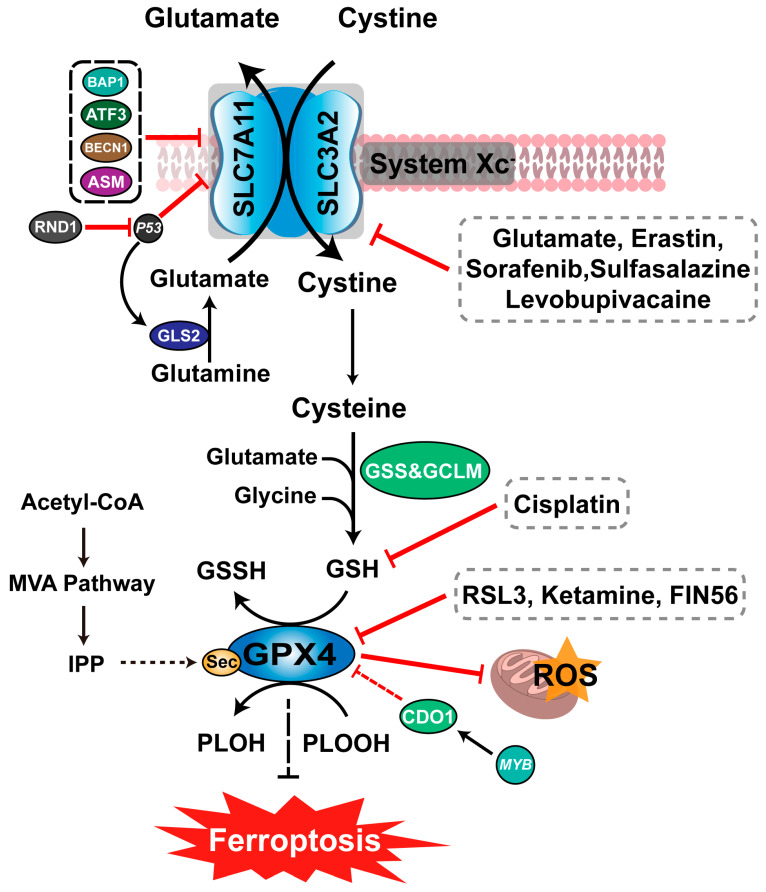
Ferroptosis and amino acid metabolism. The System Xc^−^-GPX4 pathway inhibits ROS and lipid peroxidation and thus ferroptosis. Glutamate, Erastin, Sulfasalazine, Soraferib and Levobupivacaine induce ferroptosis by inhibiting System Xc^−^, while Katamine, FIN56 and RSL3, induce ferroptosis by inhibiting GPX4. In addition, the MVA pathway can also indirectly promote GPX4 antioxidant function through the intermediate product IPP.

**Figure 2 viruses-15-02373-f002:**
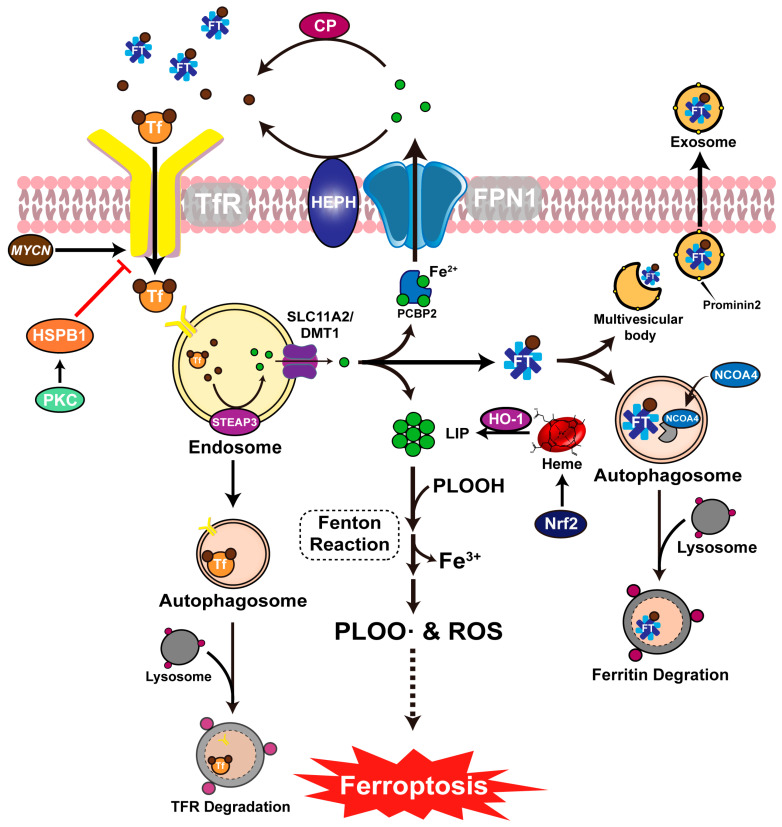
Ferroptosis and iron metabolism. Extracellular Fe^3+^ binding Tf and entry mediated by TFR. *MYCN* gene upregulates TFR expression while protein kinase C downregulates TFR by activating HSPB1. Fe^3+^ is reduced to Fe^2+^ in endosomes by STEAP3, and the reduced Fe^2+^ is released into the cytoplasm via SCL11A2. The endosome then forms an autophagosome and degrades TFR through lysosomes. Part of Fe^2+^ is stored in ferritin after reduction, and part of Fe^2+^ is transferred extracellularly via FPN1 with the assistance of the chaperone PCBP2, and this part of Fe^2+^ oxidizes to Fe^3+^ under the membrane proteins HEPH or serum ceruloplasmin. Another part of the free or weakly bound Fe^2+^ will form LIP, which induces a Fenton reaction in the presence of lipid peroxides, leading to ferroptosis. Intracellular ferritins are secreted outside the cell by prominin2, while a part of ferritins form autophagosomes and are degraded by lysosomes.

**Figure 3 viruses-15-02373-f003:**
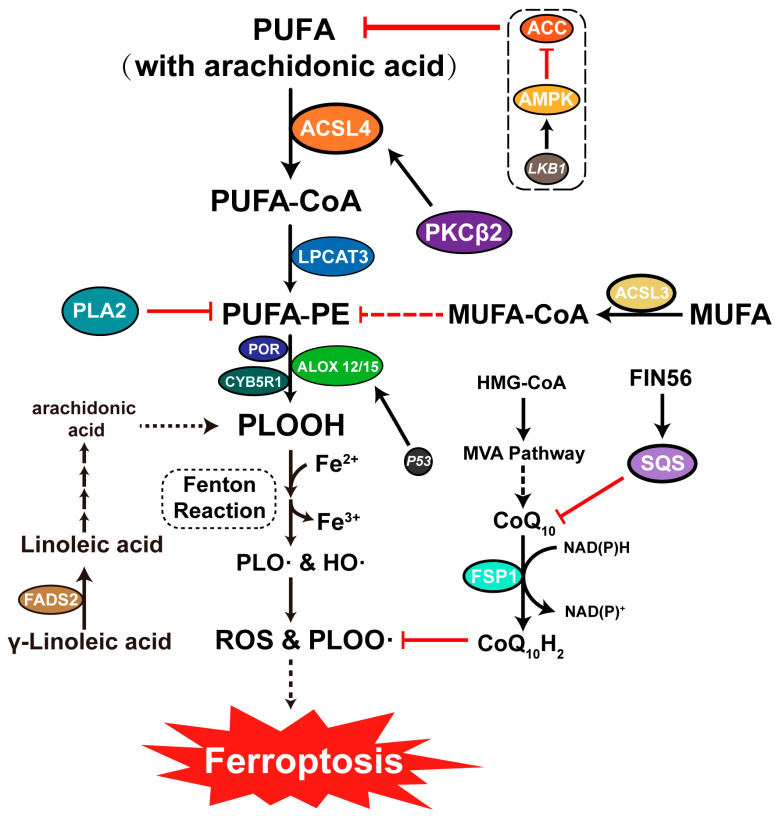
Ferroptosis and lipid metabolism. PUFA is bound to CoA by ACSL4, forming PUFA-CoA. Subsequently, PUFA-CoA is transferred to the cell membrane, where it is catalyzed by LPCAT3. ALOX12 or ALOX15 oxidizes PEs to form PLOOH, which induces ferroptosis through a Fenton reaction in the presence of Fe^2+^. In addition, ferroptosis is inhibited by eliminating PLOOH from the cell membrane through the production of CoQ_10_H_2_ in the MVA pathway. However, the production of CoQ10H2 can be inhibited by FIN56.

**Figure 4 viruses-15-02373-f004:**
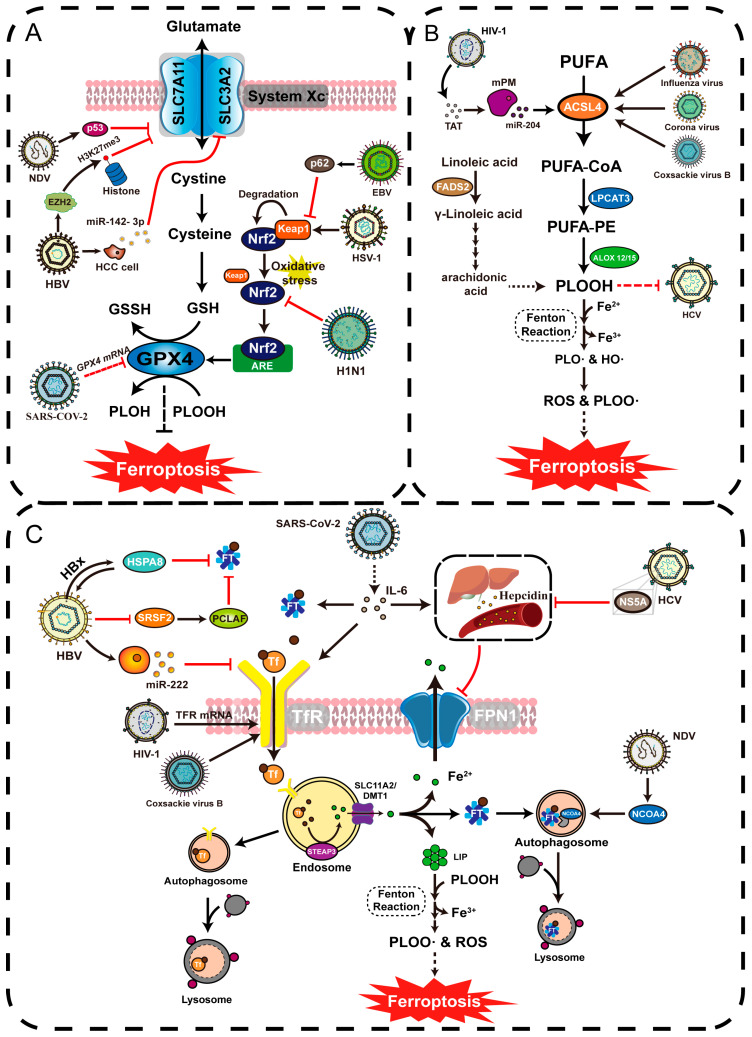
(**A**) HBV, SARS-CoV-2, H1N1, HSV-1, EBV and NDV affect the host immune response and increase the sensitivity of cells to ferroptosis by targeting the host System Xc^−^-GSH-GPX4 axis, thereby altering the redox level in the host cells and serving the reproduction. (**B**) Influenza virus, COVID-19 and CVB promote self-reproduction by recruiting ACSL4, while HIV-1 upregulates ACSL4 expression through exosomes secreted by infected cells, and overexpression of ACSL4 leads to lipid deposition and increases the risk of ferroptosis. In HCV, FADS2-dependent lipid peroxidation inhibits replicative enzymes to limit HCV replication and cause persistent intracellular infection. (**C**) SARS-CoV-2, HCV, NDV, HIV and CVB3 regulate the levels of ferritin, transferrin, and hepcidin in host cells by causing intracellular iron overload, which induces ferroptosis and promotes viral proliferation. Contrarily, HBV reduces intracellular iron levels by inhibiting TFR and FT thus inhibiting ferroptosis and promoting the development of hepatocellular carcinoma and liver fibrosis.

**Table 1 viruses-15-02373-t001:** The inducer and inhibitor in ferroptosis.

Fuction	Target	Molecule	Chemical Structure	Action Site
Inducer	System Xc^−^	Erastin [11]	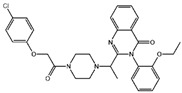	Erastin prevents inhibition of microtubule proteins on VDAC, causing an increase in mitochondrial metabolism and promoting ROS. Erastin reduces cystine input by inhibiting System Xc^−^.
		Sulfasalazine (SAS) [12]	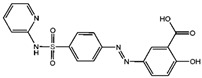	SSZ reduces cystine input by inhibiting System Xc^−^.
		Sorafenib [13]	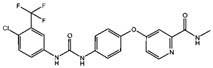	Sorafenib binds to System Xc^−^ and inhibits its activity.
		Glutamate [7]	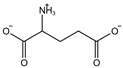	High extracellular concentrations of Glutamate affect glutamate and cystine transport and thus System Xc^−^.
		Levobupivacaine [14]	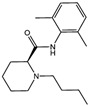	Levobupivacaine inhibits SLC7A11 or GPX4 by upregulating miR-489-3p to induced ferroptosis.
	GPX4	(1S,3R)-RSL3 [15]	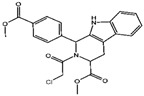	RSL3 directly inactivates GPX4 by covalently binding to Sec46 of GPX4.
		FIN56 [16]	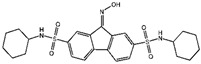	FIN56 promotes the degradation of GPX4 and leads to CoQ_10_ depletion by binding to squalene synthase.
		Ketamine [17,18]	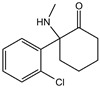	Ketamine inhibits GPX4 expression via lncRNA PVT1 or GPX4 transcript levels.
	GSH	Cisplatin (Cis) [19,20]	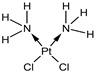	Cis binds to GSH forming a complex thereby reducing GSH concentration.
	ROS or iron	Dihydroartemisinin (DHA) [21,22,23]	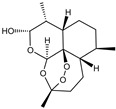	DHA induces ferritin degradation, upregulates TfR to increase free iron levels promoting ROS deposition and downregulates System Xc^−^.
Inducer	ROS or iron	Artesunate (ART) [24,25]	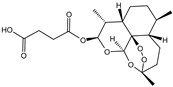	ART acts similarly to DHA in promoting ferritin degradation and ROS Accumulation.
		FINO_2_ [26]	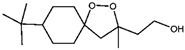	FINO_2_ undergoes the Fenton reaction with Fe^2+^ and oxidizes it to Fe^3+^ leading to the production of free radicals.
		Vitamin C [27,28]	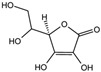	Vitamin C induces ferritin degradation, leading to free iron release.
		Sodium arsenite (NaAsO_2_) [29]	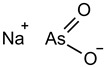	NaAsO_2_ induces ROS release by inducing mitochondrial damage and enhances ferritin degradation thereby promoting Fe^2+^ release.
Inhibitor	ROS or iron	DFO (Deferoxamine) [30]	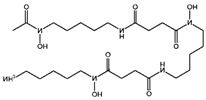	DFO binds to intracellular iron and prevents its participation in the Fenton reaction that generates ROS.
		Fer-1 (Ferrostatin-1) [7]	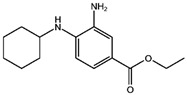	Fer-1 inhibits lipid peroxidation by trapping free radicals and stabilizing lipid peroxides. And it can it binds to free Fe^2+^ to reduce mitochondrial Fe^2+^ content.
		VE (Vitamin E) [31]	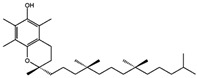	Vitamin E forms hydroquinone substances and subsequently interact with iron in the active center of the LOXs enzyme.
		liproxstatin-1 (Lip-1) [32,33]	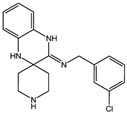	Lip-1 stays inside the lipid bilayer and removes ROS and also activates Nrf2 to restore GPX4 levels.
	LOX	Baicalein [34,35]	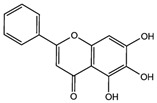	Baicalein chelates iron and inhibits 12/15-LOX activity.
		zileuton [36]	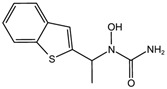	Zileuton inhibits 5-LOX activity.

Ferroptosis is involved in most pathogenic microbial infections and diseases caused by tissue and organ damage. Especially in early stages of pathogenic microbial infections when ferroptosis is manipulated to promote the microorganisms’ replication and the clinical use of ferroptosis inhibitors such as DFO and Fer-1 are often used to slow down the disease. Vitamin E is also used as an antioxidant to protect cells from ferroptosis. As for cancer treatment, ferroptosis inducer is often used clinically to induce tumor cell ferroptosis and thus inhibit growth, such as Piperazine Erastin, SAS, Sorafenib, (1S,3R)-RSL3, and Dihydroartemisinin, etc., all of which are effective in the anti-tumor treatment. However, these drugs currently have many potential risks and side effects and are not used on all tumor cells. Further research is still needed to obtain safer and more effective ferroptosis-targeted drugs in clinic.

## Data Availability

The authors confirm that all the data used in the article supporting this study are available within the article.

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
