# Peer review of "Molecular Mechanisms of Ferroptosis and Its Role in Viral Pathogenesis"

_viruses, 2023, doi:10.3390/v15122373_

Round 1

Reviewer 1 Report

Comments and Suggestions for Authors

The review titled “Molecular mechanisms of ferroptosis and its role in viral pathogenesis” comprehensively summarized the mechanism of ferroptosis and associated signaling pathways. The authors also discussed the manipulation of the ferroptosis pathway by multiple viruses and its significance in viral pathogenesis. The present article is well-written and well-structured with proper citations. However, the conclusion section is very short. Authors should provide an in-depth discussion regarding the exploitation of the ferroptosis mechanism by multiple viruses and its significance for developing targeted therapeutic implications.

Comments on the Quality of English Language

Minor editing of English language required

Author Response

Reviewer #1:

  1. Point: The review titled “Molecular mechanisms of ferroptosis and its role in viral pathogenesis” comprehensively summarized the mechanism of ferroptosis and associated signaling pathways. The authors also discussed the manipulation of the ferroptosis pathway by multiple viruses and its significance in viral pathogenesis. The present article is well-written and well-structured with proper citations. However, the conclusion section is very short. Authors should provide an in-depth discussion regarding the exploitation of the ferroptosis mechanism by multiple viruses and its significance for developing targeted therapeutic implications.

Response: Much grateful for your kind comment. We have added a paragraph in the conclusion section in which we summarized several ways the viruses regulate ferroptosis, as well as potential applied of ferroptosis inhibitors or inducers to modulate viruses  influence and proliferation.

  1. Point: Comments on the Quality of English Language

Minor editing of English language required

Response: Much grateful for your kind comment. The language of the paper has been thoroughly checked and we have rectified several instances of singular and plural errors.

We want to express our great appreciation to you the Editorial Board and Reviewers for their constructive comments and suggestions on our paper. We are looking forward to hearing from you soon. Best regards.

Yours sincerely,

Kai Kang

College of Coastal Agriculture, Guangdong Ocean University

Zhanjiang, Guangdong 524088, China.

E-mail address: kangkai610@126.com

Reviewer 2 Report

Comments and Suggestions for Authors

The manuscript entitled Molecular mechanisms of ferroptosis and its role in viral pathogenesis aesthetically summary and map the roles of ferroptosis in viral infections contribute to a further understanding of ferroptosis in relation to viruses infection. However, there are still some shortcomings in this review, for which the following relevant suggestions are made.

1. The summary of inhibitors and activators of ferroptosis is incomplete in the section 2.1. These drugs can be classified according to their mechanisms of action, and their applications in virus therapy and other related fields can be envisioned.

2. Compared with the sections 5, the content of the sections 3 and 4 is more and the section 5 is less. However, in relation to the theme of the manuscript, the content of 3, 4 and 5 should be balanced, and the discussion of the sections 5 should be increased.

3. There is a lack of content on ferroptosis in antiviral immunity in the sections 5.

4. It is recommended to add an analysis of why viral infections lead to different occurrences of GPX4 increase or suppression in the sections 5.1. Similarly, the sections 5.2 are recommended to be done like the sections 5.1.

5. The pathogenic mechanisms of viruses include virus invasion, infection, replication, and immune escape. However, the manuscript mainly discusses the landscape of ferroptosis in the sections 1 to 4. Only section 5 is related pathogenesis of viral infections, and does not to thoroughly discuss the relationship between pathogenic mechanisms and ferroptosis from virus invasion, infection, replication, and immune escape.   

5. This review lacks perspectives on ferroptosis mechanisms in relation to viral infections.

Comments on the Quality of English Language

NO

Author Response

Reviewer #2:

Comments and Suggestions for Authors

The manuscript entitled “Molecular mechanisms of ferroptosis and its role in viral pathogenesis” aesthetically summary and map the roles of ferroptosis in viral infections contribute to a further understanding of ferroptosis in relation to viruses infection. However, there are still some shortcomings in this review, for which the following relevant suggestions are made.

  1. Point: The summary of inhibitors and activators of ferroptosis is incomplete in the section 2.1. These drugs can be classified according to their mechanisms of action, and their applications in virus therapy and other related fields can be envisioned.

Response: Thank you for your precious suggestion to improve our manuscript. We have added the common ferroptosis inducer and inhibitor and provided a brief categorization of these in the table by their actions, as well as a short description of the clinical applications of inducer and inhibitor in the manuscript.

  1. Point: Compared with the sections 5, the content of the sections 3 and 4 is more and the section 5 is less. However, in relation to the theme of the manuscript, the content of 3, 4 and 5 should be balanced, and the discussion of the sections 5 should be increased.

Response: Thank you for your kind suggestion. We have added 2 paragraphs in sections 5, the titles of the paragraphs are “5.1 Transferrin Receptors Mediate Viral Entry associating with ferroptosis” and “5.2 Potential for virus-mediated ferroptosis to facilitate viral release”. In these paragraphs, we reviewed the involvement of ferroptosis-related proteins and signaling pathways in regulating viral replication at different stages of the viral replication process.

  1. Point: The pathogenic mechanisms of viruses include virus invasion, infection, replication, and immune escape. However, the manuscript mainly discusses the landscape of ferroptosis in the sections 1 to 4. Only section 5 is related pathogenesis of viral infections, and does not to thoroughly discuss the relationship between pathogenic mechanisms and ferroptosis from virus invasion, infection, replication, and immune escape.

Response: Thank you for your precious suggestion to improve our manuscript. We have added content in Section 5, following viral invasion, release, immune escape and involved iron and lipid metabolism, with the main additions of viral internalization associating with the TfR1 pathway, the potential mechanism of viral progeny release with ferroptosis, and the role of viral use of ferroptosis in relation to immune escape. The modified section can be traced in the text.

  1. Point:This review lacks perspectives on ferroptosis mechanisms in relation to viral infections.

Response: We are grateful for your comment. We have added a summary of the mechanisms about ferroptosis in viral infections in Section 6.

  1. Point:It is recommended to add an analysis of why viral infections lead to different occurrences of GPX4 increase or suppression in the sections 5.1. Similarly, the sections 5.2 are recommended to be done like the sections 5.1.

Response: Much grateful for your comment. We have briefly added an analysis of different cases of GPX4 suppression at the end of paragraphs 5.3.2 and 5.3.3.

We want to express our great appreciation to you the Editorial Board and Reviewers for their constructive comments and suggestions on our paper. We are looking forward to hearing from you soon. Best regards.

Yours sincerely,

Kai Kang

College of Coastal Agriculture, Guangdong Ocean University

Zhanjiang, Guangdong 524088, China.

E-mail address: kangkai610@126.com

Reviewer 3 Report

Comments and Suggestions for Authors

Although the subject of the manuscript concerning the developmental mechanisms through which viral infections exploit ferroptosis is very interesting, the document has many errors and the same for figures. We suggest to be extensively revised by the authors according to similar publications. English needs extensive editing because some parts of the manuscript are very confusing.

Comments on the Quality of English Language

English needs extensive editing because some parts of the manuscript are very confusing.

Author Response

Reviewer #3:

  • What is the main question addressed by the research?
  1. Point The main topic is an overview of the mechanisms underlying ferroptosis, the signaling pathways that are involved and delineate the pivotal role of ferroptosis in the pathogenesis of viral infections.

Response: thank you for your kind comment.

  • Do you consider the topic original or relevant in the field? Does it address a specific gap in the field?
  1. Point: As I commented the subject of the manuscript concerning the developmental

mechanisms through which viral infections exploit ferroptosis is very interesting. But

there is is a lack of content on ferroptosis in the different sections of the manuscript

with section 5 having a major problem.

Response: Thank you for your kind suggestion. We have added 2 paragraphs in sections 5, the titles of the paragraphs are “5.1 Transferrin Receptors Mediate Viral Entry associating with ferroptosis” and “5.2 Potential for virus-mediated ferroptosis to facilitate viral release”. In these paragraphs, we reviewed the involvement of ferroptosis-related proteins and signaling pathways in regulating viral replication at different stages of the viral replication process.

  • What does it add to the subject area compared with other published material?
  1. Point:Viruses and ferroptosis is an interesting area including a possibility to develop new antiviral strategies.

Response: thank you for your kind comment.

  • What specific improvements should the authors consider regarding the methodology? What further controls should be considered?
  1. Point: Extensive editing in english language is needed. This will improve clarification of many parts that are confusing. Overall the manuscript has a lack of perspectives on ferroptosis mechanisms in relation to viruses and there is a diffuse descriptive presentation

Response: We are grateful for your comment. We have added a summary of the mechanisms about ferroptosis in viral infections in Section 6. The article language has been completely polished by commercial editors. The certificate is provided in the system.

  • Are the conclusions consistent with the evidence and arguments presented and do they address the main question posed?
  1. Point: Sections are sufficiently organized but there is need to clarify each time the main

purpose and content of each part concerning viral infections.

Response: We are grateful for your comment. We have added a summary at the end of section 3, 5 and 6.

  • Please include any additional comments on the tables and figures.

  1. Point: Aesthetics of the different figures will be appreciated. Many spelling or typing mistakes inside figures.

Response: thank you for your comment. We checked the pictures one by one, corrected the spelling of the words and some of the content, reinserted the pictures in the revised manuscript, and uploaded the new Figures file.

We want to express our great appreciation to you the Editorial Board and Reviewers for their constructive comments and suggestions on our paper. We are looking forward to hearing from you soon. Best regards.

Yours sincerely,

Kai Kang

College of Coastal Agriculture, Guangdong Ocean University

Zhanjiang, Guangdong 524088, China.

E-mail address: kangkai610@126.com

Round 2

Reviewer 3 Report

Comments and Suggestions for Authors

I agree with all modifications of the revised manuscript.